# Combination of Immune-Related Network and Molecular Typing Analysis Defines a Three-Gene Signature for Predicting Prognosis of Triple-Negative Breast Cancer

**DOI:** 10.3390/biom12111556

**Published:** 2022-10-25

**Authors:** Jinguo Zhang, Shuaikang Pan, Chaoqiang Han, Hongwei Jin, Qingqing Sun, Jun Du, Xinghua Han

**Affiliations:** 1Department of Medical Oncology, The First Affiliated Hospital of USTC, Division of Life Science and Medicine, University of Science and Technology of China, Hefei 230052, China; 2School of Medical Oncology, Wan Nan Medical College, Wuhu 241001, China; 3School of Medical Oncology, Anhui Medical University, Hefei 230032, China; 4Department of Pathology, The First Affiliated Hospital of USTC, Division of Life Science and Medicine, University of Science and Technology of China, Hefei 230052, China

**Keywords:** triple-negative breast cancer, immune-related genes, molecular subtypes, prognosis, immunotherapy

## Abstract

Recent breakthroughs in immune checkpoint inhibitors (ICIs) have shown promise in triple-negative breast cancer (TNBC). Due to the intrinsic heterogeneity among TNBC, clinical response to ICIs varies greatly among individuals. Thus, discovering rational biomarkers to select susceptible patients for ICIs treatment is warranted. A total of 422 TNBC patients derived from The Cancer Genome Atlas (TCGA) database and Molecular Taxonomy of Breast Cancer International Consortium (METABRIC) dataset were included in this study. High immunogenic gene modules were identified using weighted gene co-expression network analysis (WGCNA). Immune-related genes (IRGs) expression patterns were generated by consensus clustering. We developed a three-gene signature named immune-related gene panel (IRGP) by Cox regression method. Afterward, the associations of IRGP with survival outcomes, infiltration of immune cells, drug sensitivity, and the response to ICIs therapy were further explored. We found five high immunogenic gene modules. Two distinct IRGclusters and IRG-related genomic clusters were identified. The IRGP was constructed based on TAPBPL, FBP1, and GPRC5C genes. TNBC patients were then subdivided into high- and low-IRGriskscore subgroups. TNBC patients with low IRGriskscore had a better survival outcome, higher infiltration of immune cells, lower TP53 mutation rate, and more benefit from ICIs treatment than high IRGriskscore patients. These findings offer novel insights into molecular subtype of TNBC and provided potential indicators for guiding ICIs treatment.

## 1. Introduction

According to estimates, female breast cancer (BC) has now displaced lung cancer as the leading cause of global cancer incidence, with an estimated 2.3 million new cases and 685,000 deaths in 2020 [1]. Among the subtypes of BC, triple-negative breast cancer (TNBC) is the most aggressive subtype that accounts for approximately 15–20% of all BC patients [2,3]. Due to the absence of ideal treatment target, TNBC is not sensitive to endocrine therapy or molecular targeted therapy [4]. Chemotherapy is one of the primary systemic treatments of TNBC, but the efficacy of conventional chemoradiotherapy differs widely and has considerable heterogeneity for different individuals. As high-throughput sequencing technologies have developed, more intrinsic subtypes of TNBC have been discovered and different subtypes have obviously different treatment guidelines and tumor biology [5]. Therefore, developments of new treatment strategies and applicable molecular subtypes of TNBC have become an urgent challenge.

In recent years, immunotherapy has emerged as an attractive and effective pillar in cancer treatment, and increasing evidence supports a key role of the immune system in determining the response to standard therapy and long-term survival in BC patients [6,7]. Recent studies suggested that TNBC was thought to be the most immunogenic BC subtype characterized by the presence of tumor neoantigens and high levels of lymphocytic infiltration [8]. According to the results of KEYNOTE-522 and IMpassion130, pembrolizumab and atezolizumab were approved by the US Food and Drug Administration (FDA) for early-stage or metastatic TNBC [9,10]. However, TNBC patients that benefit from immune checkpoint inhibitors (ICIs) remain very limited. Furthermore, other immunotherapy strategies such as adoptive T-cell transfer, vaccination, or virotherapy have not yet demonstrated meaningful clinical activity [11]. Presently, biomarkers for predicting efficacy of immune therapy have gained increased attention. Remarkably, PD-L1 expression, tumor mutation burden, and infiltrating lymphocytes (TILs) have long been proposed as reliable indicators for immune checkpoint inhibitors (ICIs) treatment [12]. As the factors that influence the efficacy of ICIs are multifaceted, many key questions remain unanswered. Therefore, exploring the expression patterns of immune-related pathways not only enhances our understanding of immune dysregulation but also provides potential opportunities for antitumor immunity in TNBC.

In this study, we aimed to investigate the aberrant immune network and explore a prognostic marker for predicting the immune microenvironment status in TNBC. Here, we comprehensively analyzed the immune network and identified immune-related subtypes based on the expression profiles of 422 TNBC patients. A gene panel to quantify the immune responses status for individual patients was established. Additionally, we linked our immune signature with immune cell-infiltrating characteristics and response to ICIs. Overall, our study systematically explored the immune-related subtypes and identified a robust indicator for predicting the prognosis of TNBC.

## 2. Materials and Methods

### 2.1. TNBC Patients and Public Datasets

RNA sequencing data of 123 TNBC patients and 113 normal samples, somatic mutation data, and corresponding clinicopathological information were obtained from The Cancer Genome Atlas (TCGA) database (www.portal.gdc.cancer.gov/projects, accessed on 2 February 2022). The mRNA expression data of Molecular Taxonomy of Breast Cancer International Consortium (METABRIC) dataset on TNBC containing 299 samples and clinical information were accessed via cBioPortal (http://www.cbioportal.org/, accessed on 10 February 2022) [13]. Next, we extracted immune-related genes (IRGs) from ImmPort database [14] (www.immport.org/shared/home, accessed on 15 February 2022), InnateDB database [15] (www.innateDBdb.com/, accessed on 15 February 2022) and GSEA database (http://www.gsea-msigdb.org/gsea/index.jsp, accessed on 20 February 2022). After removal of duplicate genes, 4327 immune-related genes were acquired for further analysis. Patients without the required clinical data were excluded from all analyses. Batch effects of individual datasets were adjusted using ComBat.

### 2.2. Differential Expression and Enrichment Analysis

The limma algorithm [16] was used to select the differentially expressed genes (DEGs) and differentially expressed immune-related genes (DEIGs) between TNBC sample and normal tissues in TCGA-TNBC cohort. The screening threshold was log2 fold change > 1 and false discovery rate (FDR) < 0.05. Enrichment analysis for Gene Ontology (GO) and Kyoto Encyclopedia of Genes and Genomes (KEGG) pathways was performed using the R package ClusterProfiler [17]. According to the *p* value, the TOP 8 significant biological terms were visualized as a Circos plot.

### 2.3. Identification of High Immunogenicity Modules by WGCNA

We used the weighted gene co-expression network analysis (WGCNA) to identify high immunogenicity modules in TNBC [18]. Firstly, similarity matrix was constructed by calculating the Pearson correlation coefficient between gene pairs based on the DEIGs expression data. Then, we used soft threshold of β = 8 to convert the similarity matrix into an adjacency matrix, which was able to maintain the scale-free topology and sufficient node connectivity. A topological overlap matrix (TOM) was created from the adjacency matrix to describe the degree of association between genes. The dynamic pruning tree was designated to identify the modules using 1-TOM as the distance measure. Finally, we identified 7 modules by setting the merging threshold function at 0.25. The modules with significantly higher IRGs expression patterns in TNBC tumor tissues were selected for subsequent analysis. The expression of IRGs in these high immunogenicity modules was also extracted for further analysis.

### 2.4. Consensus Molecular Clustering of IRGs and Gene Set Variation Analysis (GSVA)

According to the expression of IRGs, unsupervised clustering analysis was used to identify diverse IRGs expression patterns. We categorized TNBC patients into different molecular subtypes for subsequent analysis. The optimal number of clusters was established using the cophenetic coefficient, dispersion, and silhouette. All subtypes identification was conducted by utilizing the R package “ConsensusClusterPlus” with 1000 times repetitions [19]. In order to understand the biological processes involved in different IRGs patterns, we performed GSVA enrichment analysis using the R package “GSVA” [20]. The gene set of “c2.cp.kegg.v7.2.symbols” and “h.all.v7.4.symbols” were obtained from the MsigDB database (http://www.gsea-msigdb.org/gsea/downloads.jsp, accessed on 17 March 2022). A list of the top 20 enriched pathways was selected with adjusted *p* < 0:05. To estimate the abundance of 23 immune cell infiltrations in TNBC, the single-sample gene-set enrichment analysis (ssGSEA) algorithm was performed as described before [21]. Gene sets for each immune cell type that we used were reported, previously [22]. The enrichment scores were limited to the range of 0 to 1.

### 2.5. Generation of IRGs Gene Signature and Functional Enrichment Analysis

To screen the IRG-related DEGs among different IRGs clusters, we utilized the “limma” method with the filtering criteria of adjusted *p* < 0.05. The univariate Cox regression model was applied to assess the prognostic value of IRG-related DEGs. Then, the IRGs gene clusters were also achieved with consensus clustering algorithm. To identify functional enrichment of IRG-related DEGs, annotation analysis of GO and KEGG pathway was carried out with R package “ggplot2”, “clusterProfiler”, “org.Hs.eg.db”, and “enrichplot”. The top 30 biological terms were used with *p* < 0.05.

### 2.6. Construction and Validation of the Immune-Related Gene Panel (IRGP)

We combined TCGA-TNBC cohort and METABRIC-TNBC cohort to create a final meta-cohort. To establish a prognostic gene panel, we used the least absolute shrinkage and selection operator (LASSO) method to evaluate the associations between IRGs expressions and overall survival (OS) [23]. In this way, each TNBC sample could be calculated with a risk score by the following formula: (coef 1 × expression of gene 1) + (coef 2 × expression of gene 2)+. +(coef n×expression of gene n). According to the median of risk score, patients with TNBC were divided into high- and low-risk subgroups for subsequent study. Survival analysis in high- and low-risk subgroups was conducted by the R “survival” and “survminer” packages. R package “survivalROC” was used to produce receiver operating characteristic (ROC) curves. Analyses of dimensionality reduction were performed using principal component analysis (PCA) and t-distributed stochastic neighbor embedding (t-SNE).

### 2.7. Comprehensive Analysis of Immune Characteristics and ICIs Therapy in Different IRGP Subgroups

To identify immune properties of TNBC samples, mRNA expression data were calculated by the CIBERSORT algorithm to estimate the abundance of 22 types of immune cells [24]. The associations between the abundance of 22 types of immune cells and clinicopathological factors were presented as a landscape map. To clarify the immune function in high- and low-risk subgroups, we identified the IRGP signatures enrichment level as ssGSEA score. For the effects of IRGP signature on gene mutation, a mutation profile was visualized using the R/Bioconductor ‘maftools’ package in two IRGP subgroups [25]. In calculating the correlations between IRGP score and PD-1, PD-L1 and CTLA-4 expression, Spearman’s correlation analysis was performed. To predict chemotherapeutic response, the “pRophetic” R package was implemented to analyze the expression data of each patient [26]. All drug response data were collected from the Genomics of Drug Sensitivity in Cancer (GDPS) database (https://www.cancerrxgene.org/, accessed on 25 March 2022) [27]. Immunophenoscore (IPS) data were retrieved from online database (https://tcia.at/, accessed on 25 March 2022) for TCGA-TNBC [22]. Analysis of IPS between low and high IRGs riskscore was performed according to the status of CTLA-4 and PD-1 expression.

### 2.8. Detection of the Expression of IRGP by qRT-PCR and Human Protein Atlas (HPA) Database

MCF-10A human mammary epithelial cells were cultured in mammary epithelial basal medium supplemented with mammary epithelial cell growth kit. MDA-MB-231, BT549 and SKBR-3 were cultured with DMEM (GIBCO) media supplemented with 10% FBS. MCF-7 was grown in RPMI-1640 media (GIBCO) supplemented with 10% FBS. Total RNA was extracted and reverse transcribed using Thermo Scientific RevertAid MM (Thermo Fisher Scientific, Waltham, MA, USA). Quantitative RT-PCR was conducted with Roche LightCycler 96 using Universal SYBR Green Fast qPCR Mix (Abclonal, Woburn, MA, USA). The normalization reference gene was GAPDH. The sequences of primers in this study were as follows: Human FBP1 forward, CGCGCACCTCTATGGCATT; Human FBP1 reverse, TTCTTCTGACACGAGAACACAC; Human TAPBPL forward, CTGCCTGGCTCTATCTGGAG; Human TAPBPL reverse, CCTTGGAAATCGGTGAAGTCC; Human GPRC5C forward, CCTGTACTACAACCTGTGTGAC; Human GPRC5C reverse, TGAGCACAAACGTGGTGACA. The protein level of FBP1, TAPBPL, and GPRC5C was detected by using the Human Protein Atlas (HPA) database (https://www.proteinatlas.org/, accessed on 25 March 2022).

### 2.9. Statistical Analysis

In this study, all statistical analyses were performed using R-4.1.1(Vienna, Austria). Continuous variables were compared between the two groups by using the independent *t*-test. The Chi-square test was used to test categorical data. A Kaplan–Meier survival analysis based on univariate log-rank test was performed. Statistical significance was defined as a *p* value below 0.05.

## 3. Results

### 3.1. Identification of High Immunogenicity Modules in TNBC by WGCNA

Schematic summary of the proposed model is presented as Figure 1. First, we performed differential mRNAs expression analysis between 123 TNBC tumors and 113 normal controls in the TCGA database. As a result, 6927 differentially expressed mRNAs were screened, of which 3584 genes were upregulated and 3343 genes were downregulated in the TNBC samples compared with normal samples (Appendix A). Taking the intersection of DEGs with IRGs, 1139 differentially expressed IRGs were obtained, of which 690 genes were upregulated and 449 were downregulated in TNBC samples (Figure 2A). Functional annotation and pathway analysis revealed that these genes were involved in the process of T cell activation, negative regulation of immune system, and chemokine signaling pathway (Figure 2B,C). The top 20 terms of KEGG pathways and GO annotation were shown in Appendix A. To obtain the high immunogenicity modules in TNBC, WGCNA analysis was performed based on the expression of DEIGs. The outliers’ sample was removed by sample clustering (Appendix A). The optimal soft-thresholding power of 8 was selected determined by the scale-free network (Appendix A). A hierarchical cluster analysis was applied to detect co-expression clusters and 7 co-expression modules were found using dynamic tree-cutting methods (Figure 2D). Among the 7 co-expression modules, DEIGs in 5 modules (magenta, green, pink, turquoise, gray) showed higher expression in TNBC tumors than normal (Figure 2E). Since these IRGs from the 5 modules were more enrichment in TNBC samples, indicating TNBC patients might show a higher expression of these IRGs. Finally, the expression of IRGs in the 5 high immunogenicity modules was extracted for further analysis.

### 3.2. Construction of Distinct IRGs Expression Patterns in TNBC

We combined the TCGA-TNBC and METABRIC-TNBC cohorts to generate a final meta-cohort. Consensus clustering analysis was applied to classify TNBC samples into different IRGs patterns and two IRGs clusters were identified (Figure 3A and Appendix A). We termed it as IRGcluster A or B and survival analysis showed that TNBC patients with IRGcluster B had a better overall survival (OS) than IRGcluster A (Figure 3B). ssGSEA scores were calculated to quantify the abundances of 23 immune-infiltrating cells in different IRGs clusters. IRGcluster B subtype presented a higher abundance of activated CD8 T cell, activated dendritic cell, natural killer cells, and so on (Figure 3C). Heatmap demonstrated that the expression of IRGs was higher in IRGcluster B subtype compared to IRGcluster A (Figure 3D). GSVA was conducted to explore the biological process in two IRGs clusters. As shown in Figure 3E, IRGcluster-B was enriched in KRAS signaling, TNFA signaling, and interferon gamma response. Furthermore, immune-related pathways including natural killer cell mediated cytotoxicity, T cell receptor signaling, and antigen processing and presentation were highly enriched in IRGcluster-B subgroup (Figure 3F). Together these results demonstrated that ICGcluster B was a high immunogenicity TNBC subtype characterized by abundant immune cells infiltration and activated immune-related pathways.

### 3.3. Generation of IRGs Gene Signatures and Construction of the Immune-Related Gene Panel (IRGP)

We further screened the DEGs between two IRGclusters. In total, 530 DEGs were recognized, and functional enrichment analysis demonstrated that these genes were involved in leukocyte mediate immunity, major histocompatibility complex (MHC) protein complex, and various immune processes (Appendix A). Then, we performed univariate cox regression analysis and 266 prognostic IRGs were associated with OS in TNBC (Appendix A). Unsupervised consensus clustering further classified TNBC into two IRG gene clusters (Figure 4A). The Kaplan–Meier curve illustrated that IRG-genecluster B was correlated with better OS compared with IRG-genecluster A (Figure 4B). To construct an immune-related gene panel (IRGP) for predicting the OS of TNBC patients, we applied LASSO regression based on the expression of 266 prognostic IRGs and a three-gene signature was identified (Figure 4C,D). The risk score was calculated by the formula IRGP = expression level of TAPBPL × (−0.41) + expression level of FBP1 × (−0.22) + expression level of GPRC5C × (0.26). Interestingly, a lower risk score was observed in IRGcluster B and IRG-genecluster B (Figure 4E,F). The alluvial diagram illustrated that IRGcluster B had a high overlap with the IRG-genecluster B subtype, which showed a lower IRGriskscore with a favorable survival prognosis (Figure 4G).

### 3.4. Validation of the Capacity of IRGP 

Based on the median IRGriskscore, TNBC cohort was divided into low-risk and high-risk subgroup. TNBC patients were randomly classified as two datasets: the training cohort (N = 211) and testing cohort (N = 210). Kaplan–Meier survival curves showed that TNBC patients with lower IRGriskscore had a longer survival time than those with higher IRGriskscore in both the training and testing cohort (Figure 5A,B). ROC curves with AUC at 1 year, 3 years, and 5 years also demonstrated the moderate accuracy of our IRGP (Figure 5C,D). Distribution of risk scores between high risk and low risk groups was shown in Figure 5E,F. The scatter plots demonstrated that an increase of deaths was accompanied by increasing risk score in both the training cohort and testing cohort (Figure 5G,H). As shown in Figure 5I,J, the expression of TAPBPL and FBP1 was lower in high-risk group, while a higher expression of GPRC5C was observed in high-risk group.

### 3.5. The mRNA and Protein Level of IRGP in BC

We next detected the mRNA level of FBP1, TAPBPL, and GPRC5C by RT-qPCR. As Figure 6A shows, FBP1 was significantly overexpression in MDA-MB-231, MCF7 and SKBR-3 cells. TAPBPL was significantly upregulated in BT549 and SKBR-3 cells (Figure 6B). High expression of GPRC5C was found in MDA-MB-231 and SKBR-3 cells compared to MCF-10A cells (Figure 6C). We also evaluated the protein expression and subcellular location using HPA database. The expression of FBP1 and TAPBPL was significantly high in BC tissues, and moderate expression of GPRC5C was found both in breast normal tissue and cancer tissue (Figure 6D–I). These proteins are largely localized to the cytosol and the membrane.

### 3.6. Prognostic Value and Genomic Features in Different IRGP Subgroups

PCA and t-SNE analysis showed that IRGP subgroups presented obvious segregation in training cohort (Figure 7A,B). A similar result was also observed in the testing cohort (Figure 7C,D). To investigate whether the effect of IRGP on OS was an independent factor for TNBC patients, univariate and multivariate Cox analyses for variables including age, tumor stage, tumor size, node status, and IRGriskscore were performed. Univariable Cox regression analysis revealed that IRGriskscore was a high-risk factor for TNBC (HR = 2.188, 95% CI (1.360, 3.522), *p* < 0.001, Figure 7E). Multivariate Cox analysis indicated that IRGP could serve as an independent prognostic factor (HR = 2.216, 95% CI (1.384, 3.548), *p* < 0.001, Figure 7F). Then, we explored the genomic features in different IRGP subgroups of TCGA-TNBC cohort. The waterfall plot revealed that the low IRGriskscore group presented a decreased mutation burden compared with the high IRGriskscore group (Figure 7G,H). Among the top 20 most significant mutated gene, TP53 (84% vs. 75%) had higher mutation rates in the high IRGriskscore group.

### 3.7. Immune Characteristics of Different IRGP Subgroups

Immune checkpoints gene expression was often considered as indicators for ICIs treatment. Firstly, the expression of PD-1, PD-L1, and CTLA-4 was significantly higher in low IRGriskscore group compared to high IRGriskscore group (Figure 8A–C). IRGriskscore also had a significant negative correlation with the expression of PD-1, PD-L1, and CTLA-4 (Figure 8D–F). Immune cells landscape suggested that the proportion of CD8 T cells, B naïve cell, plasma cells, M1 macrophages, and T helper cells were more abundant in the low IRGriskscore subgroup, while M2 macrophages were more abundant in the high IRGriskscore subgroup (Figure 8G). Correlation analysis further suggested that IRGriskscore presented a positive correlation with M0 macrophages, M2 macrophages and T regulatory cells (Figure 8H–J). However, a negative correlation was found between IRGriskscore and activated NK cells, CD4 memory T cells, CD8 cells, and M1 macrophages (Figure 8K–N). We also found TNBC patients with high IRGriskscore tended to have a higher level of tumor mutation burden (Appendix A), but this difference was not statistically significant. All in all, TNBC patients with low IRGriskscore had a higher abundance with tumor-infiltrating immune cells.

### 3.8. Relationship of IRGP Subgroups with Immune Subtypes and IPS

As for the correlation between IRGP and clinicopathological parameters, we found from Figure 9A that our IRGriskscore was significantly correlated with patients’ tumor size. Thorsson et al. reported the immune landscape of pan-cancer of TCGA database and defined six immune subtypes [28]. As shown in Figure 9B, TNBC patients mainly belonged to the subtype of C1 (Wound Healing) and C2 (IFN-γ Dominant). More C2 subtypes were enriched in low IRGriskscore group (*p* = 0.001, chi-square test). Our IRGP signature contains three immune genes: FBP1, GPRC5C, and TAPBPL. As in Figure 9C, FBP1 had a strong correlation with infiltration of resting dendritic cell, resting mast cells and T helper cells. The expression of GPRC5C was strongly associated with the infiltration of M2 macrophages, resting mast cells and activated CD4 memory T cells. Furthermore, a strong positive correlation was seen between the expression of TAPBPL and M1 macrophages, activated NK cells, activated CD4 memory T cells, and CD8 T cells. We then used IPS to assess the potential clinical efficacy of ICIs in different IRGP groups. The higher IPS prediction score indicated a better response to ICIs therapy. There was no significant difference for PD-1 and CTLA-4 negative expression patients (Figure 9D). However, low IRGriskscore might be a robust indicator for ICIs therapy in TCGA-TNBC cohort for patients with PD-1 or CTLA-4 positive expression (Figure 9E–G). Although TNBC patients with high IRGriskscore might not be the most appropriate population for ICIs therapy, our drug sensitivity analysis demonstrated that high IRGriskscore subgroup was less sensitivity to PARP inhibitors (ABT.888 and AZD.2281), doxorubicin, gemcitabine, and methotrexate (Appendix A).

## 4. Discussion

Although women with TNBC make up 12–17% of all breast cancer cases, it is clinically recognized as having the worst outcomes among all BC subtypes [29]. With rapid growth in a multi-omics approach, many newly discovered types of TNBC have been identified [30,31]. Genomically, TNBC is heterogeneous and characterized by distinct molecular subtypes that lead to different biological behavior and treatment responses [32]. Recent advances in antitumor immunotherapy by targeting the PD-1/PD-L1 checkpoint have led to significant clinical improvements in TNBC and HER2 positive breast cancers [33]. The clinical efficacy of cancer immunotherapy is associated with individual tumor immune microenvironment and “hot” tumor tend to have a better ICIs response than “cold” one [34]. Thus, identification of TNBC subtypes with highly immunogenic will facilitate a rational design of anti-tumor immunotherapy. Recently, the involvement of gene signatures has been described in several studies based on multi-omics data. Yan et al. developed a risk autophagy-related prediction model that can predict the survival status of TNBC patients [35]. Yang et al. constructed a risk scoring system to predict the immune activity and potential therapeutic response of TNBC [36]. Our previous work revealed an eight immune-related panel in the immunomodulatory subtype of TNBC [37]. In the previous study, we screened an eight immune-related and prognostic gene signature based on WGCNA. However, our current study combined the WGCNA algorithm with TNBC molecular typing analysis in two TNBC cohorts. We identified two distinct IRGclusters and developed a three genes panel to predict the prognosis of TNBC. This study aimed to explore the role of immune-related genes network in a systems biology manner and investigated the immunotherapeutic value of the immune-related signature in TNBC.

Various molecular subtypes of TNBC have been identified, including basal-like 1 (BL1), basal-like 2 (BL2), immunomodulatory (IM), mesenchymal (M), mesenchymal stem-like (MSL), and luminal androgen receptor (LAR) [38]. The IM subtype manifests high levels of immune signal transduction pathways and immune-related genes [39]. The expression levels of PD-L1, PD-1, and CTLA4 were reported to be higher in IM subtype than other subtypes [31]. The FUTURE trial showed that anti PD-1 strategy could be a promising treatment for refractory metastatic TNBCs with IM subtype [40]. To find the high immunogenic subtype, we first screened the differentially expressed immune-related genes. Then, we identified the high expression ICGs modules by WGCNA and further categorized TNBC into two subtypes based on the high immunogenicity TNBC modules. Further analysis revealed that IRGcluster-B had a superior survival compared to IRGcluster-A. Abundant immune cell infiltration was observed in IRGcluster-B patients with elevated immune genes expression levels. Thus, TNBC patients with IRGcluster-B subtype might belong to immune “hot” tumor and be more likely to benefit from immunotherapy.

To investigate the major difference at molecular level between these two subtypes, we further constructed IRG-related genomic clusters and two gene clusters were identified. Interestingly, IRG-genecluster B has a better survival outcome than IRG-genecluster A, which was similar to IRGs cluster patterns. To evaluate different IRGs expression for individual patients, a quantification scoring model termed IRGriskscore was adopted by LASSO algorithm. Previous studies have also attempted to construct an immune-related gene prognostic index to distinguish the molecular and immune characteristics in various solid tumors [41,42,43]. For the associations of IRGriskscore with IRGcluster and IRG-genecluster, we found the IRGcluster-B and IRG-genecluster B subtype exhibited a lower IRGriskscore. To verify the accuracy and efficacy of our proposed model, we then assessed the survival outcomes, diagnostic value, risk curve, and univariate and multivariate analyses in training and validation dataset. As a result, there was little variation in the training and validation dataset, suggesting our model had certain universality, efficacy, and accuracy. Overall, the TNBC patients with low IRGriskscore was associated with improved prognosis.

Here, TAPBPL, FBP1, and GPRC5C were applied to build our IRGs signature. These genes were identified as prognostic immune signatures in TNBC for the first time. Our results of RT-qPCR and IHC demonstrated that TAPBPL, FBP1, and GPRC5C were high expression in BC cell lines and tissues. TAP binding protein like (TAPBPL), as a novel T cell co-inhibitory factor, was reported to localize on chromosome position 12p13.3 near an MHC paralogous locus [44]. Study showed that TAPBPL was expressed on the surface of antigen presenting cells (APCs) and played a critical role in negatively controlling T-cell functions [45]. Targeting TAPBPL-mediated inhibitory pathway might be effective strategy for the patients who were resistant to PD-1/PD-L1 or CTLA-4 antitumor therapy [45]. The various immune microenvironment components contribute to maintaining the equilibrium state of body [46]. As the immune cell infiltration increases, negative immune regulation factors may be activated. Thus, these negative immune regulation factors including PD-L1 or TAPBPL are accompanied by higher level of immune cells infiltration. As a rate-limiting enzyme in gluconeogenesis, fructose-1,6-bisphosphatase (FBP1) controls the rate of the reaction and works as an important tumor suppressor in human malignancies [47]. Previous study demonstrated that overexpression of FBP1 conferred sensitivity to cisplatin via modulating STAT3 in ovarian cancer [48]. Another study reported that FBP1 inhibited tumorigenesis of cholangiocarcinoma partly via Wnt/β-catenin pathway [49]. The study by Cong et al. revealed that aberrant expression of FBP1 in NK cells resulted in their dysfunction by inhibiting glycolysis and impairing viability [50]. FBP1 knockdown decreased the migration and invasion of MDA-MB-231 cells and enhanced the sensitivity of TNBC cells to cisplatin [51]. To date, the role of FBP1 in anti-tumor immune response is not well-understood. GPRC5C belongs to the type 3G protein-coupled receptor family [52]. Knockdown experiments of GPRC5C potentially regulated the proliferation or migration of MCF-7 cells [53]. It was reported that GPRC5C was found in inrenal proximal tubules and participated in regulating renal acid-base homeostasis [54]. In pancreatic β-cells, downregulation of GPRC5C resulted in a decreased of glucose-stimulated insulin release and cAMP content [55]. However, the function of GPRC5C in antitumor immunity remain unclear.

Recent evidence demonstrated that infiltrating lymphocytes (TILs) was observed in HER2+ BC and TNBC patients [56]. Therefore, BC has ceased to be considered as an immunological quiescent tumor type. Previous research indicated that abundance of TILs in the tumor microenvironment was related to better response to ICI therapies [57,58]. In the present study, patients with low IRGriskscore displayed a higher level of TILs compared to high IRGriskscore subgroup. IRGriskscore was negatively associated with infiltrating of activated NK, CD4, and CD8 cells, suggesting that our IRGriskscore might be a good predictive marker for ICI therapies. Currently, atezolizumab and pembrolizumab were approved in combination with chemotherapy among patients with unresectable locally or metastatic TNBC expressing PD-L1 [59,60]. Our results showed that high level of PD1, PD-L1 and CTLA4 was observed in TNBC patients with low IRGriskscore, which strongly implicated that TNBC patients with low IRGriskscore might be suitable for anti-tumor immunotherapy. For the effect of IRGP signature on gene mutation rate in TNBC, our findings demonstrated that patients with low IRGriskscore had unusually low mutation rates compared to high IRGriskscore group, specifically for TP53. Previous studies reported that TP53 mutation status were associated with depressed immune signatures and the efficiency of ICIs therapy in solid tumors [61,62,63]

The study by Thorsson et al. identified six immune subtypes based on the differences in lymphocyte signatures across TCGA pan-cancer [28]. The subtype of C1 (wound healing) had increased expression of angiogenic genes and C2 (IFN-γ dominant) subtype showed a strong CD8 signal and the highest M1/M2 macrophage polarization [63]. In our analysis, we found TCGA-TNBC cohort was primarily categorized as C1 and C2 subtype. Most low IRGriskscore TNBC patients belong to C2 subtype, suggestive of high lymphocyte expression signature. Based on previous studies, IPS have been identified as useful biomarkers for predicting patient response to immunotherapy [22]. IPS was able to quantify the determinants of tumor immunogenicity and had predictive value in cancer patients treated with the CTLA-4 and PD-1 blockers [22]. In this study, we also evaluated the predictive value of the TNBC IRGriskscore using IPS. Low IRGriskscore had the potential to be a robust predictor for PD-1 or CTLA-4 therapy and its predictive value depended on the PD-1 or CTLA-4 expression. Meanwhile, drug sensitivity analysis revealed that TNBC patients with high IRGriskscore was associated with high IC50 of PARP inhibitors, doxorubicin, gemcitabine and methotrexate, which will probably guide clinical practice for chemotherapeutic agents’ decision-making.

To our knowledge, we first identified immune-related subtypes of TNBC using WGCNA combined with consensus clustering. There are several notable limitations to our study. First, our immune signature was constructed based on TCGA and METABRIC database. Thus, prospective cohort of TNBC patients receiving ICIs therapy are needed to validate our findings. Second, the effect of our gene signature on immune infiltration should be experimentally verification in the future. Thirdly, comparison of predictive value between different models also should be conducted.

## 5. Conclusions

In conclusion, we identified the DEIGs in TNBC and found five high immunogenicity modules by WGCNA. We developed two distinct IRGclusters, two IRG-related genomic clusters, and IRGriskscore to assess the characteristics of immune cells infiltration. The three-gene panel was strongly correlated with patient’s prognosis, the abundances of immune cell infiltration, immunophenotyping of TNBC, and chemotherapy and immunotherapy sensitivity. Our study offered novel insights into the molecular subtype of TNBC and provided potential indicators for guiding ICIs treatment.

## Figures and Tables

**Figure 1 biomolecules-12-01556-f001:**
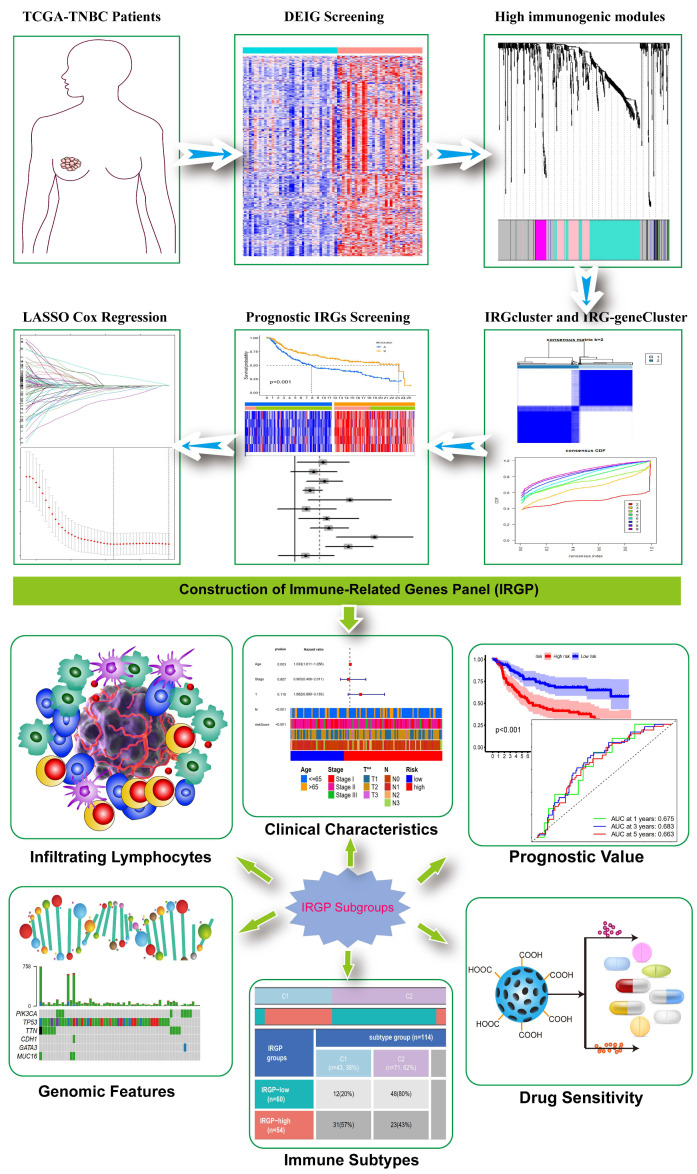
The flowchart of study design of current study.

**Figure 2 biomolecules-12-01556-f002:**
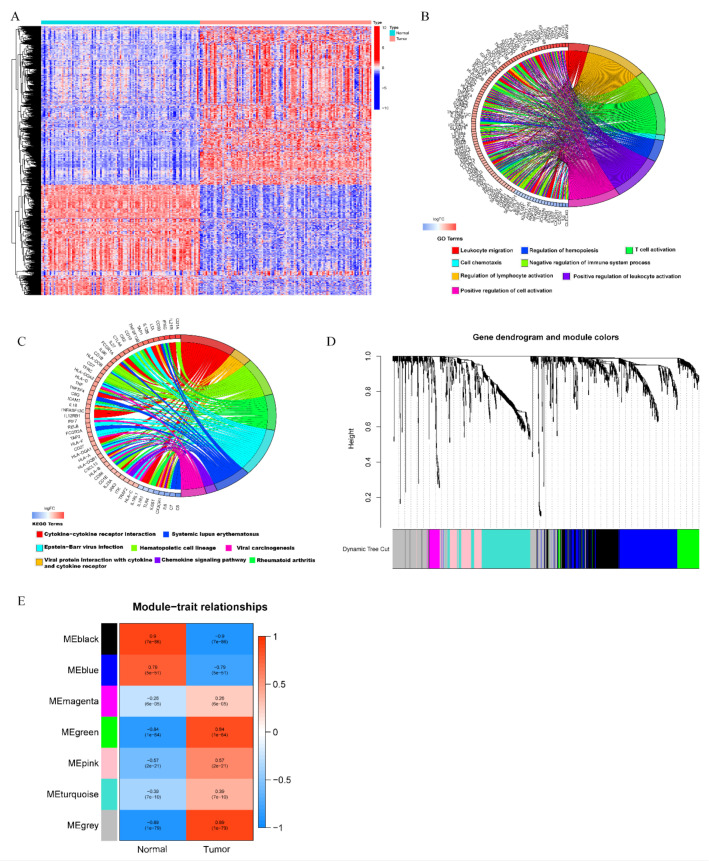
Identification of differentially expressed immune-related genes (DEIGs) and high immunogenicity modules in TNBC. (**A**) Heatmap of differentially expressed immune-related genes between TNBC tumor samples (red) and normal samples (blue) using the “limma” method. (**B**) Circos graph for GO enrichment of DEIGs. (**C**) Circos graph for KEGG pathways analysis of DEIGs. (**D**) Clustering dendrogram of DEIGs. Each short vertical line corresponds to a gene. The branches are modules of highly interconnected groups of gene expression. Seven modules were identified, and the lower panel shows colors designated for each module. (**E**) The module–trait relationships between the identified modules and clinical status (normal and tumor). Rows are module eigengene (ME) regards to each module, and the columns indicate traits. Red represents high adjacency and blue represents low adjacency. The upper number in each cell indicates the correlation coefficient of each module in the trait, and the lower number is the corresponding *p*-value.

**Figure 3 biomolecules-12-01556-f003:**
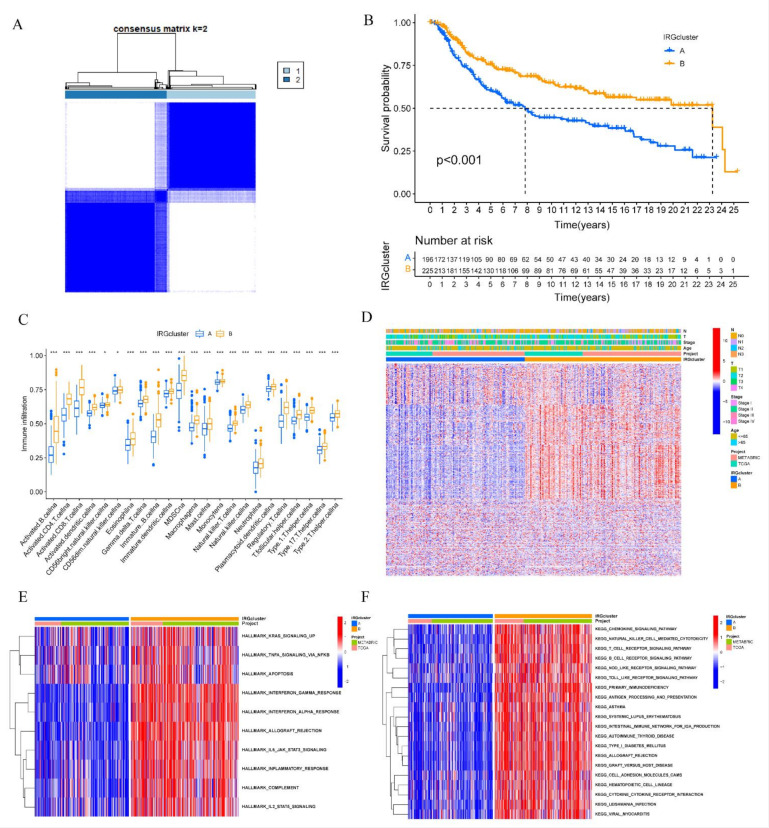
Identification of distinct IRGs subtypes in TNBC. (**A**) Consensus clustering matrix for k = 2 based on IRGs expression. (**B**) Kaplan–Meier curves of OS for two TNBC subtypes. The numbers of patients in IRGcluster-A and IRGcluster-B are 196 and 225, respectively. (**C**) The abundance of infiltrating immune cells in two IRGs subtypes. The lines in the boxes were the median value. The top and bottom ends of the boxes indicated interquartile range of values (* *p* < 0.05; *** *p* < 0.001). (**D**) Unsupervised clustering of IRGs expression to divide TNBC patients into two IRGs subtypes. The IRG clusters, projects, age, stage, tumor size, and node status were used as patient annotations. (**E**,**F**) Heat map showed the GSVA score of hallmark signature and KEGG pathways in different IRGcluster. Red means activated pathways and blue means inhibited pathways.

**Figure 4 biomolecules-12-01556-f004:**
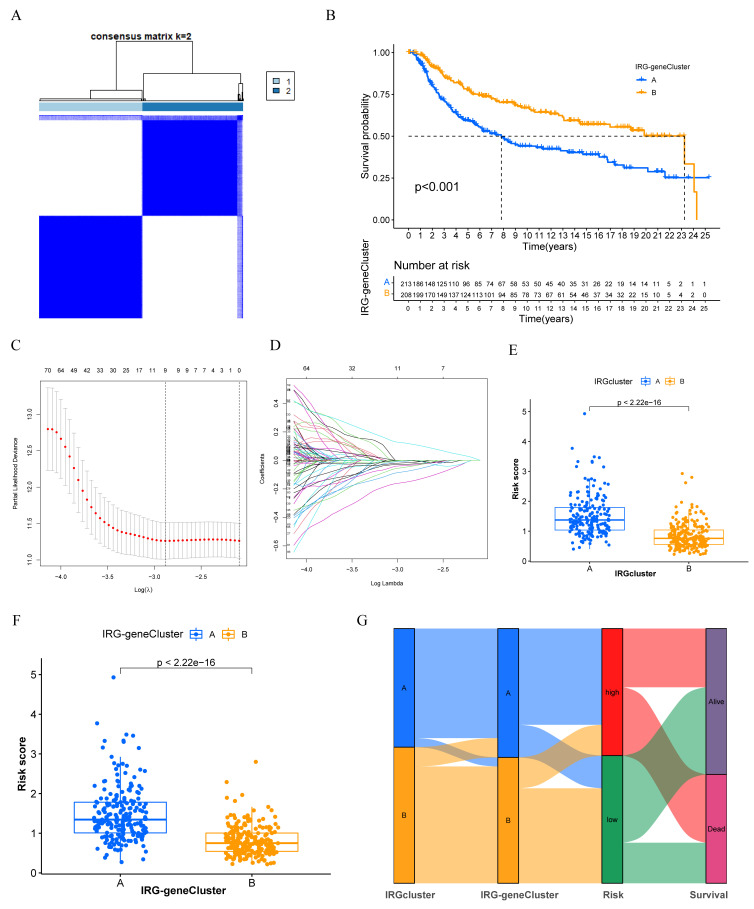
Construction of IRGs geneCluster and the immune-related gene panel (IRGP). (**A**) The identification of IRG-geneCluster by consensus clustering matrix for k = 2. (**B**) Survival curves of the IRG-geneCluster were plotted by the Kaplan–Meier plotter. The numbers of patients in IRG-geneCluster A and IRG-geneCluster B subtypes are 213 and 208. (**C**) Elucidation for LASSO coefficient profiles of prognostic IRGs. (**D**) The least absolute shrinkage was performed and construction of selection operator (LASSO) regression model. (**E**) The correlation between IRGs risk score and IRGs subtypes. (**F**) The correlation between IRGs risk score and IRG-geneCluster. (**G**) Alluvial diagram of IRGs clusters in groups with different IRG-geneCluster, IRGs risk score, and OS.

**Figure 5 biomolecules-12-01556-f005:**
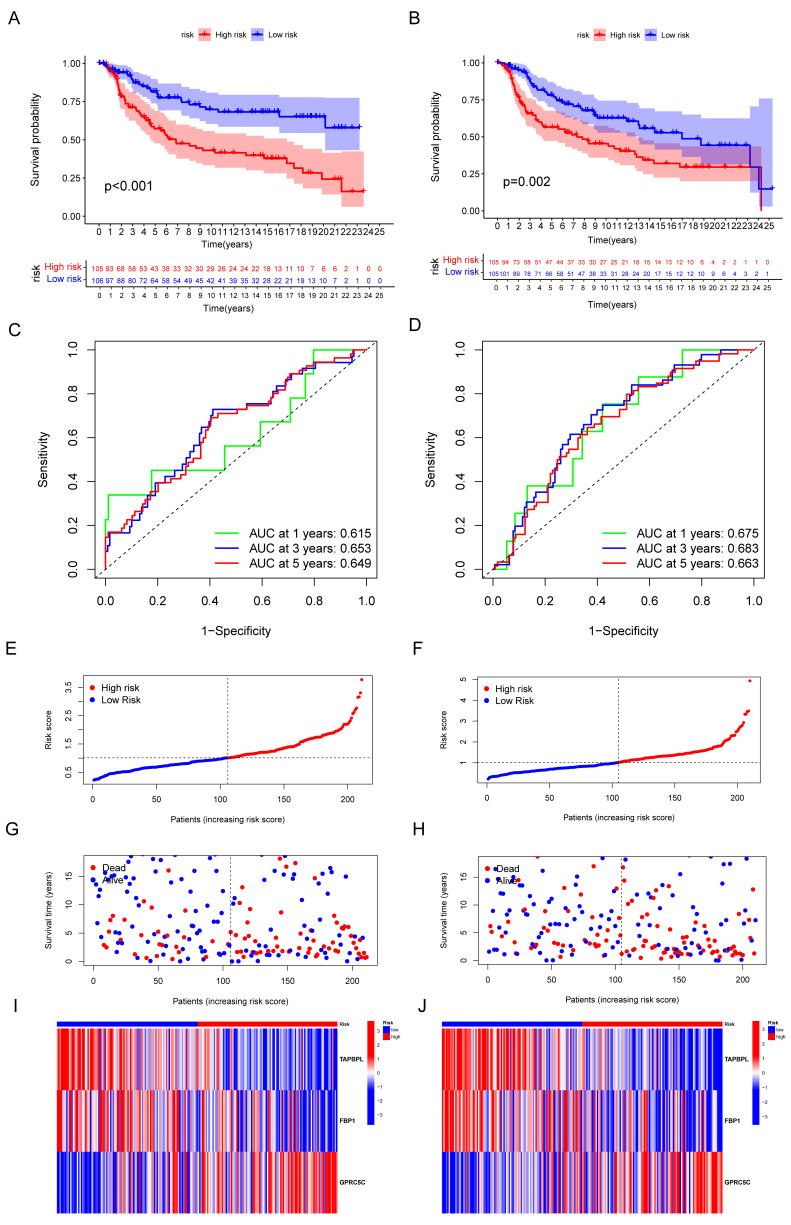
Validation of the efficiency of IRGP. (**A**,**B**) Kaplan–Meier curves for OS of TNBC patients according to the risk stratification in the training cohort and testing cohort. (**C**,**D**) ROC analysis for OS prediction at 1 year, 3 years, and 5 years in TNBC patients in the training cohort and testing cohort. (**E**,**F**) Distribution of risk score for the training group and testing group. (**G**,**H**) Distribution of survival time of patients in the training and testing groups. (I,J) Heatmap depicting the expression of TAPBPL, FBP1, and GPRC5C between high-risk group and the low–risk group in training and testing groups.

**Figure 6 biomolecules-12-01556-f006:**
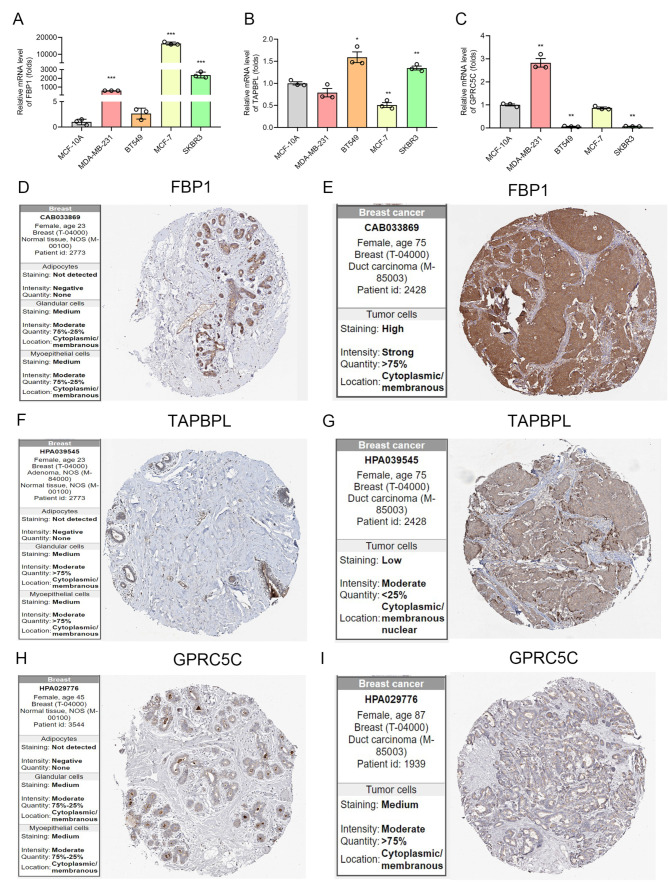
The mRNA and Protein level of IRGP in BC. (**A**–**C**) The mRNA levels of three immune-related genes in BC cell lines MDA-MB-231, BT549, MCF-7, and SKBR-3 cells compared with normal breast epithelial cells MCF-10A cells by qRT-PCR. * *p* < 0.05, ** *p* < 0.01, *** *p* < 0.001. (**D**–**I**) Representative immunohistochemistry (IHC) staining of FBP1, TAPBPL, and GPRC5C in breast carcinoma and normal breast tissues.

**Figure 7 biomolecules-12-01556-f007:**
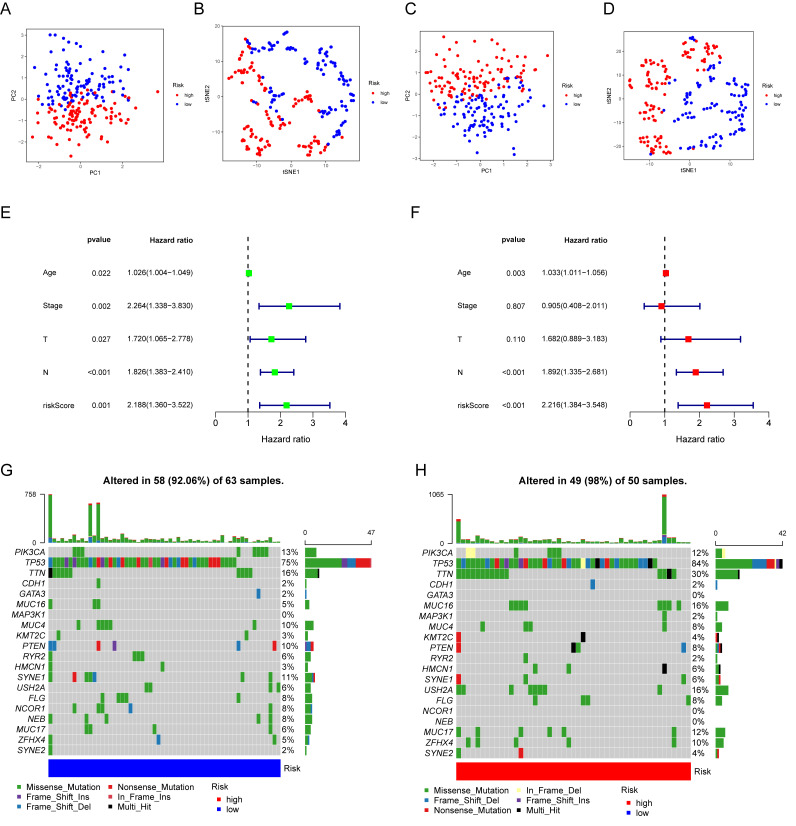
Prognostic value and genomic features in different IRGP subgroups. (**A**,**B**) PCA analysis and t-SNE analysis of TNBC patients in different IRGP subgroups for the training cohort. (**C**,**D**) PCA analysis and t-SNE analysis of TNBC patients based on IRGP model for the testing cohort. (**E**) Forest plot for IRGP model score and clinical features in TNBC patients by univariate analyses. (F) The multivariate Cox forest plot of IRGP model score and clinical characteristics. (**G**,**H**) The waterfall plot showing the somatic mutation rate in low IRGP riskscore and high IRGP riskscore. Each column indicates an individual sample. The upper bar plot is the tumor mutation burden for an individual sample. The right histogram generalizes the percentage of each variant type. The number on the right indicates the mutation frequency for each gene.

**Figure 8 biomolecules-12-01556-f008:**
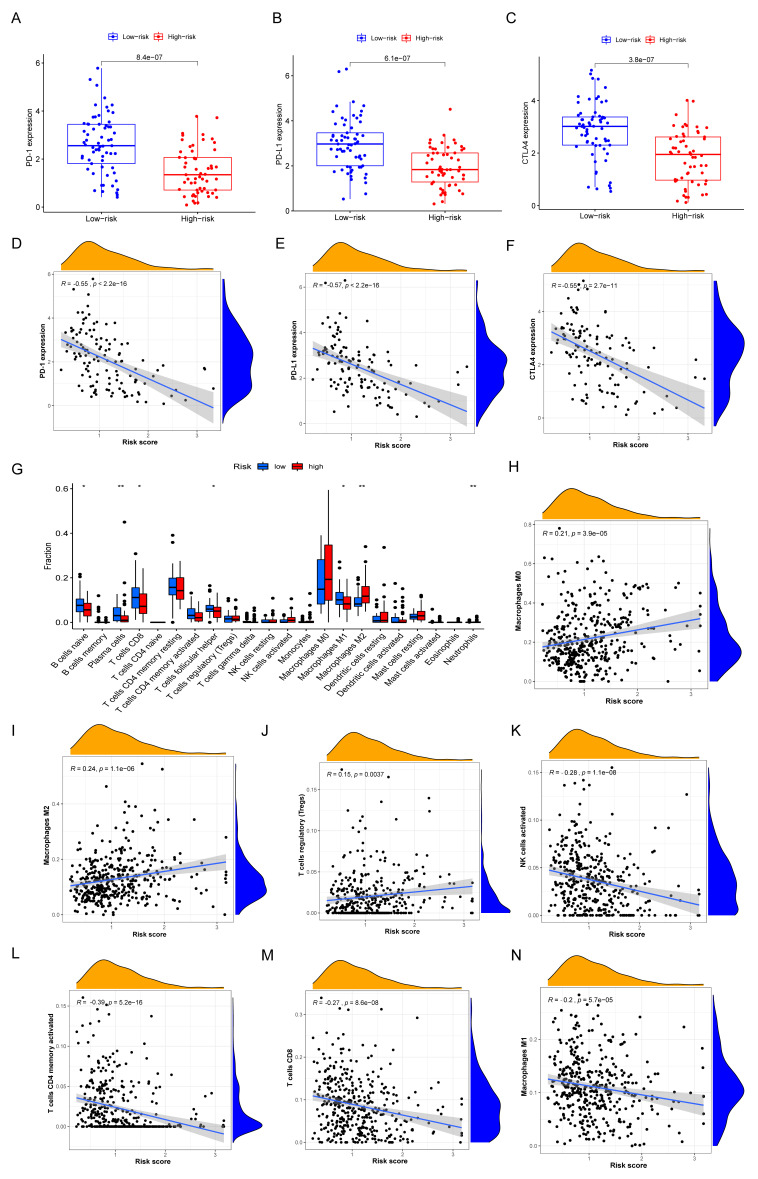
The immune characteristics of different IRGP subgroups. (**A**–**C**) The expression of PD–1, PD-L1, and CTLA–4 in high– and low–riskscore TNBC groups. (**D**–**F**) IRGP signature negatively related to the expression of PD–1, PD–L1, and CTLA–4. (**G**) The abundance of each infiltrating immune cells in different IRGP subgroups. The lines in the boxes were the median value. The top and bottom ends of the boxes showed interquartile range of values (* *p* < 0.05; ** *p* < 0.01). (**H**–**J**) IRGP signature positively related to infiltrating of M0 macrophages, M2 macrophages, and T regulatory cells. (**K**–**N**) IRGP signature negatively correlated with infiltrating of activated NK cells, CD4 memory T cells, CD8 cells and M1 macrophages.

**Figure 9 biomolecules-12-01556-f009:**
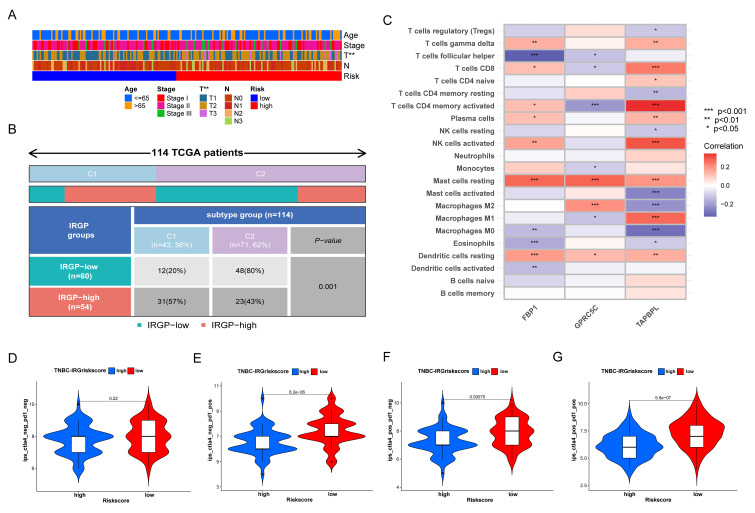
Predictive value of IRGP model in immune subtypes and IPS. (**A**) Heatmap showing the relationship between IRGP model and clinicopathological parameter, ** *p* < 0.01. (**B**) Heatmap showing the relationship between IRGP model and immune subtypes. (**C**) Heatmap showing the correlation of FBP1, GPRC5C and TAPBPLwith the infiltrating of 22 immune cells. (**D**–**G**) The relative distribution of IPS identified by the status of CTLA-4 or PD-1 was compared between high IRGriskscore versus low IRGriskscore in TCGA-TNBC cohort.

## Data Availability

We declare that all the data in this article are authentic, valid, and available for use on reasonable request. Publicly available datasets or databases were analyzed in the present study. This data can be found here: TCGA database (https://portal.gdc.cancer.gov, accessed on 2 February 2022) and cBioPortal database (http://www.cbioportal.org/, accessed on 10 February 2022). Jinguo-Zhang can be contacted (zhangjg2022@ustc.edu.cn) regarding the availability of data and materials.

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
