# Peer review of "Combination of Immune-Related Network and Molecular Typing Analysis Defines a Three-Gene Signature for Predicting Prognosis of Triple-Negative Breast Cancer"

_biomolecules, 2022, doi:10.3390/biom12111556_

Round 1

Reviewer 1 Report

This study explores the differentially expressed immune-related genes (DEIGs) in triple-negative breast cancer (TNBC) extracted from TCGA and METABRIC cohorts and develops a three-gene panel for patient stratification with correlation to patient’s prognosis and immunophenotype. There are several critical issues needed to be addressed before this manuscript can be accepted for publication.

1.          The authors conducted many different types of analyses but the logic is unclear. Are all these analyses needed and appropriate? It is recommended to have a flowchart (Figure 9) first to express the logic and strategy for these analyses. However, the initial screening for DEIGs was based on TCGA cohort with 123 (Figure 1), not 422 (Figure 2), TNBC patients (Figure 9).

2.          The authors have reported an 8-immune gene panel for TNBC (Front Oncol 2020, 10, 1787). It is required to compare and discuss the results of both studies.

3.          Figure 1B, 1C: please provide justifications for the selection of these GO and KEGG pathways compared to other ignored pathways (e.g. a table showing the p-values of top 20 pathways).

4.          Figure 1F: please explain the reason for the use of WGCNA to identify the high immunogenicity modules in TNBC. What’s the purpose/significance to discover the “co-expression modules”? Why these “co-expression modules” can represent high immunogenicity? How many DEIGs in the 5 co-expression modules are high-expression in TNBC? What is the relationship between the DEIGs in the 5 co-expression modules and those in the selected GO and KEGG pathways (Figure 1B, 1C)? How many and which of them are overlapped? Figure 1D can be put into supplementary Figure S2.

5.          Figure 2: the analyzed base (TCGA+METABRIC) is different from that in Figure 1 (TCGA). However, in the last sentence of section 3.1 (lines 183-184, for Figure 1), the DEIGs in the 5 “high immunogenicity modules”, which were identified from TCGA cohort, were used for downstream analyses (TCGA+METABRIC). Is it appropriate? Why does not include METABRIC cohort at the initial analysis? Or why does not use all 1139 DEIGs (Figure 1A), instead of that in the 5 “high immunogenicity modules”, for clustering IRG subtype?

6.          Please clarify the numbers of DEIGs used for “IRGcluster” (for Figure 2) and “IRG-genecluster” (for Figure 3).

7.          It is strange that LASSO identified FBP1, GPRC5C, and TAPBPL when compared to the results of Kaplan-Meier survival analysis (Table S1). How are the three genes identified but they are not so prognostic? What are their significance and implication based on their functions in TNBC? For example, TAPBPL was reported as a negative regulator of T cells; however, the authors did not explain why TAPBPL played a good prognostic value in TNBC (HR < 1, Table S1)?

8.          In Figure 3E, 3F, the TNBC patients have been stratified into high- and low-risk scores according to the constructed “Immune-Related Gene Panel” (IRGP). It is strange to re-classified “Based on the median IRGriskscore” in section 3.4. So, what classification method was applied to other analyses, such as Cox regression for HR and others (Figures 6-8)?

9.          Figure 4A-B: validation should use another cohort, not by re-allocating the same patients into two subgroups, which does not make sense.

10.      Figure 5: the expressions of FBP1, GPRC5C, and TAPBPL in cancer cell lines cannot reflect the situation in patient’s cancer tissues. This does not provide any support to their prognostic role in patients (high-risk versus low-risk).

11.      There are many redundant contexts between Results and Discussion sections.

Minor points:

1.        In Discussion: “high IRGriskscore was associated with high sensitivity of PARP inhibitors, doxorubicin, gemcitabine and methotrexate” (lines 438-439), where “high sensitivity” should be “high IC50” or “low sensitivity”.

2.          In Title: defines

3.          The letters in many figures are too small.

Reviewer 2 Report

In this manuscript, 422 TNBC patients were included to identify immune-related subtypes. consensus clustering and Cox regression method were used to develop Immune-related genes (IRGs) and immune-related gene panel (IRGP). Afterward, the associations of IRGP with survival outcomes, infiltration of immune cells, drug sensitivity and the response to ICIs therapy were further explored. Although most of the data used in this paper come from databases, it still offers novel insights into molecular subtype of TNBC and provided potential indicators for guiding ICIs treatment. This paper may be reconsidered after major revision. The detailed comments are as follows. 

1. Whether the data from TCGA and METABRIC databases has been normalized to remove the influence of batch effects is not clarified in the text. 

2. In "2.9. Statistical analysis" in the text, it is mentioned that the wilcoxon test is used for the statistical comparison of the TIDE score, but the analysis and calculation of the tide are not found in the subsequent text and pictures. 

3. Although two cohorts were randomly divided in the TCGA and METABRIC database and obtained a certain validation effect, the ROC curve results of the model seem to indicate that its validation effectiveness is limited, so robust external cohort validation is still indispensable. We recommend that the authors add a new external independent validation cohort to more robustly validate the survival probability and the predictive power of the model. 

4. The gene pane and theme in this paper are the characteristics and differences of immune subtypes of TNBC, while the quantitative RT-PCR experiments are compared with cell lines such as MCF-10A, MDA-MB-231, BT549, MCF-7 and SKBR3. And the three genes are not specifically expressed in TNBC cell lines. Therefore, a further explanation is required for the purpose and conclusion of the comparison among them. In addition, the author selected the relevant IHC data of normal breast tissue and breast cancer tissue in the HPA database. Please clarify the comparison intention and the analysis of the results in this study. 

5. TMB is closely related to immune response, is there any difference in TMB between the two subtypes in this paper? 

6. The details in the text need to be further checked, such as "P<0:05" and "P<0:05, P<0:001" in lines 111, 121, 218, etc., The subtitle label format of "3.2" is inconsistent with others. Please review the full text carefully and revise it. 

7. The reference format in the manuscript does not conform to the requirements of this journal. It is recommended to further review the reference format and revise as required.

 8. It is recommended to use fewer abbreviations to improve readability and do not add abbreviations if they appear only once. When abbreviations appear for the first time, give the full name. For example, "APC", "MHC", etc. Please review this article carefully and revise it. 

9. In Figures 2B and 3B, the total number of cohorts is 421 after merging the TCGA and METABRIC databases. However, the abstract and methods section shows that including 422 patients in this study. If there are other exclusion criteria, please specify. 

10. There is occlusion in Figure 6A, please re-upload after adjustment.

Round 2

Reviewer 1 Report

The authors have answered my questions. However, please verify the case number  “422 TNBC” patients in revised Figure 1.

Author Response

The authors have answered my questions. However, please verify the case number “422 TNBC” patients in revised Figure 1.

Response: Thank you for your advice. We have modified the Figure 1 according to your suggestion.

Reviewer 2 Report

The authors have made most revisions to the previous problems, but there are still some problems that need to be corrected. The details are as follows:

1. In Supplementary Figure S6, what is the ordinate "tmbation" meaning?

2.Although TMB expression differences between the two groups were added in the article, there were no statistical significance, and it is recommended to add a description in the main text.

3. The name of the uploaded images are not in the same order as those in the main text. Please carefully review and modify the image before uploading again.

4. In the main text, the author showed that “The optimal soft thresholding power of 6 was selected determined by the scale-free network (Supplementary Figure S3)”. However, in the supplementary materials section, “Supplementary Figure. S3 Network topology for different soft-thresholding powers. A soft-thresholding power of 8 was selected to achieve maximal model fit.” Please confirm the picture details and explain the reasons for selection.

5. According to the results of your supplementary Table S4, the 2 does not seem to be the best option. Please give a reasonable explanation for the 2-classification choice.

6. Supplementary Figure S6, S7, Supplementary Table S1, S2 are not consistent with the description in the supplementary materials section, please be sure to check the details before uploading!

Round 3

Reviewer 1 Report

The revised version is acceptable for publication.

Author Response

Thank you very much for your comments and professional advice. These opinions help to improve academic rigor of our article.

Reviewer 2 Report

In the manuscript and the file upload section, the supplementary files uploaded is inconsistent, and the image naming displayed in the main text is inconsistent with the uploaded image file name. For example, the Figure 9 you uploaded is actually Figure 1 in the manuscript. To avoid ambiguity, please rename the picture and upload it again after confirming the supplementary files and the upload location.

Author Response

We're sorry that we cannot find any inconsistence between figures and main text. It might be the difference system of authors and reviewer. We cannot change the order of figures that we uploaded firstly. All the order of figures was based on the lasted revised manuscript.